# Prostate-Specific Membrane Antigen Positron Emission Tomography Oncological Applications beyond Prostate Cancer in Comparison to Other Radiopharmaceuticals

**DOI:** 10.3390/diagnostics14101002

**Published:** 2024-05-13

**Authors:** Alberto Miceli, Virginia Liberini, Giovanna Pepe, Francesco Dondi, Antonio Vento, Lorenzo Jonghi Lavarini, Greta Celesti, Maria Gazzilli, Francesca Serani, Priscilla Guglielmo, Ambra Buschiazzo, Rossella Filice, Pierpaolo Alongi, Riccardo Laudicella, Giulia Santo

**Affiliations:** 1Nuclear Medicine Unit, Azienda Ospedaliero-Universitaria SS. Antonio e Biagio e Cesare Arrigo, 15121 Alessandria, Italy; alberto.miceli@ospedale.al.it; 2Nuclear Medicine Unit, ASO S.Croce e Carle Cuneo, 12100 Cuneo, Italy; liberini.v@ospedale.cuneo.it (V.L.); buschiazzo.a@ospedale.cuneo.it (A.B.); 3Nuclear Medicine Unit, Fondazione IRCCS Policlinico San Matteo—Pavia V.le Camillo Golgi, 27100 Pavia, Italy; gi.pepe@smatteo.pv.it; 4Nuclear Medicine Unit, ASST Spedali Civili di Brescia, 25123 Brescia, Italy; francesco.dondi@unibs.it; 5Nuclear Medicine Unit, ASP 1—P.O. San Giovanni di Dio, 92100 Agrigento, Italy; antvento@hotmail.it; 6Nuclear Medicine Unit, Fondazione IRCCS San Gerardo dei Tintori, 20900 Monza, Italy; l.jonghilavarini@campus.unimib.it; 7Nuclear Medicine Unit, Department of Biomedical and Dental Sciences and of Morpho-Functional Imaging, University of Messina, 98122 Messina, Italy; clsgrt94c44f158d@studenti.unime.it (G.C.); rlaudicella@unime.it (R.L.); 8Nuclear Medicine Unit, ASL Bari—Di Venere Bari, 70131 Bari, Italy; maria.gazzilli@asl.bari.it; 9Nuclear Medicine Unit, Presidio Ospedaliero Santo Spirito, 65124 Pescara, Italy; francesca.serani@asl.pe.it; 10Nuclear Medicine Unit, Veneto Institute of Oncology IOV-IRCCS, 35128 Padua, Italy; priscilla.guglielmo@iov.veneto.it; 11Nuclear Medicine Unit, University Hospital “Paolo Giaccone”, Via del Vespro 129, 90127 Palermo, Italy; rossella.filice@policlinico.pa.it; 12Nuclear Medicine Unit, A.R.N.A.S. Ospedali Civico, Di Cristina e Benfratelli, 90127 Palermo, Italy; pierpaolo.alongi@arnascivico.it; 13Nuclear Medicine Unit, Department of Experimental and Clinical Medicine, “Magna Graecia” University of Catanzaro, 88100 Catanzaro, Italy

**Keywords:** prostate-specific membrane antigen, oncology, positron emission tomography, FDG, unconventional PSMA PET

## Abstract

Background: Prostate-specific membrane antigen (PSMA) is a type II transmembrane glycoprotein overexpressed on the surface of tumor cells in most of the patients affected by prostate adenocarcinoma (PCa). However, PSMA expression has also been demonstrated in the endothelial cells of newly formed vessels of various solid tumors, suggesting a role for PSMA in neoangiogenesis. In this scenario, gallium-68 (^68^Ga) or fluoro-18 (^18^F)-labeled PSMA positron emission tomography (PET) may play a role in tumors other than PCa, generally evaluated employing other radiopharmaceuticals targeting different pathways. This review aims to investigate the detection rate of PSMA-PET compared to other radiopharmaceuticals (especially [^18^F]FDG) in non-prostate tumors to identify patients who may benefit from the use of such a theragnostic agent. Methods: We performed a bibliographic search on three different databases until February 2024 using the following terms: “positron emission tomography”, “PET”, “PET/CT”, “Prostate-specific membrane antigen”, “PSMA”, “non-prostate”, “not prostate cancer”, “solid tumor”, “FDG”, “Fluorodeoxyglucose”, “FAPi”, “FET”, “MET”, “DOPA”, “choline”, “FCH”, “FES”, “DOTATOC”, “DOTANOC”, and “DOTATATE”. Only original articles edited in English with at least 10 patients were included. Results: Out of a total of 120 articles, only 25 original articles comparing PSMA with other radiotracers were included in this study. The main evidence was demonstrated in renal cell carcinoma, where PSMA showed a higher detection rate compared to [^18^F]FDG PET/CT, with implications for patient management. PSMA PET may also improve the assessment of other entities, such as gliomas, in defining regions of early neoangiogenesis. Further data are needed to evaluate the potential role of PSMA-PET in triple-negative breast cancer as a novel therapeutic vascular target. Finally, unclear applications of PSMA-PET include thyroid and gastrointestinal tumors. Conclusions: The present review shows the potential use of PSMA-labeled PET/CT in solid tumors beyond PCa, underlining its value over other radiopharmaceuticals (mainly [^18^F]FDG). Prospective clinical trials with larger sample sizes are crucial to further investigate these possible clinical applications.

## 1. Introduction

Prostate-specific antigen membrane (PSMA) is a type II transmembrane glycoprotein discovered in prostate cancer (PCa) cell lines [1]. Human PSMA, a zinc-containing metalloenzyme (750 amino acids), is encoded by the FOLH1 gene located in the short arm of chromosome 11. It is characterized by a unique three-part structure consisting of a large extracellular domain, a transmembrane portion, and an intracellular component. Totaling 707 amino acids, the extracellular region is the largest portion, containing the enzymatic domains that serve as the primary target for PSMA ligand imaging and therapy [2,3]. The role of PSMA is not completely discovered yet, but reported functions include enzymatic peptidase activity related to folate and glutamate metabolism, as well as the activation of PI3K/AKT and cAMP/PKA pathways, which are involved in cell proliferation and survival [4]. Despite the name, PSMA expression is not exclusive to prostate cells and can be found in many other tissues and pathologic conditions, including inflammation/infection, non-prostatic tumor-associated neovasculature (i.e., colon, gastric, lung, breast, glioma, adrenal, bladder, renal cell carcinoma), and non-neoplastic conditions. Incidental findings in other organs are possible as well [5,6,7,8,9,10,11]. The PSMA apical expression occurs in normal prostatic epithelial cells and is markedly amplified in PCa cells [12]. In contrast, in almost all non-prostatic solid tumors, endothelial PSMA expression is associated with tumor neovasculature (Figure 1), but it is not present in benign endothelial tissue [13,14]. The most common PET-PSMA ligands used are either ^68^Ga-labeled (i.e., [^68^Ga]Ga-PSMA-11, [^68^Ga]Ga-PSMA-617) or ^18^F-labeled (i.e., [^18^F]DCFPyL, [^18^F]PSMA-1007, and [^18^F]rhPSMA-7.3). Peculiar aspects of the physiological distribution can constitute pitfalls in image interpretation [15]. However, to date, there is no evidence of superior diagnostic accuracy or better clinical results for one specific PSMA ligand among the others [16]. Furthermore, PSMA-targeting ligands have been labeled with nuclides, such as lutetium-177 (^177^Lu) or actinium-225 (^225^Ac), with therapeutic purposes, achieving beneficial effects in advanced PCa patients with acceptable toxicity [17,18]. Supposedly, diagnostic and therapeutic PSMA ligands could also play a role in non-prostatic tumors.

The aim of this review is to investigate the detection rate of PSMA-PET compared to other radiopharmaceuticals (especially [^18^F]FDG) in non-prostatic tumors in order to identify patients with solid tumors who may benefit from this theragnostic agent. In addition, the correlation between clinicopathologic features and radiopharmaceutical uptake was also considered and discussed, highlighting differences between tracers.

## 2. Strategy Research

Three independent investigators (A.V., F.S., and L.JL.) performed a bibliographic search on PubMed, Scopus, and Google Scholar databases until February 2024. The search algorithms were different combinations of the following terms: “positron emission tomography”, “PET”, “PET/CT”, “Prostate-specific membrane antigen”, “PSMA”, “non-prostate”, “not prostate cancer”, “solid tumor”, “FDG”, “Fluorodeoxyglucose”, “FAPi”, “FET”, “MET”, “DOPA”, “choline”, “FCH”, “FES” “DOTATOC”, “DOTANOC”, and “DOTATATE”.

Only original articles edited in English studying humans with at least 10 patients were included. Case reports, editorials, and preclinical papers were not included. The additional literature was retrieved from the reference lists of all identified articles. Two independent reviewers (G.S. and F.D.) evaluated the full texts of the selected papers. After the selection, 25 articles were included in this review [19,20,21,22,23,24,25,26,27,28,29,30,31,32,33,34,35,36,37,38,39,40,41,42,43].

This systematic review was carried out using the standard methods, following PRISMA guidelines. This study’s workflow is illustrated in Figure 2.

Table 1 summarizes the main characteristics of the included studies.

## 3. Results

### 3.1. Brain Tumors

According to the European Association of Neuro-Oncology (EANO) guidelines, magnetic resonance imaging (MRI) with or without the administration of a gadolinium-based contrast agent is the modality of choice for diagnosis, response assessment, and follow up of adult patients with diffuse gliomas [44]. [^18^F]FDG PET is not the ideal radiotracer for the evaluation of Central Nervous System (CNS) tumors because of the high physiological uptake of the normal brain, which implies a reduced tumor-to-background ratio (TBR). Conversely, radiolabeled amino acids, such as ^11^C-methionine ([^11^C]MET), ^18^F-fluoroethyl-L-tyrosine ([^18^F]FET), or ^18^F-dihydroxyphenylalanine ([^18^F]F-DOPA), can help define metabolic hotspots for specific tumor tissue sampling, improve target delineation for radiotherapy planning, and differentiate treatment-related changes (TRC), such as pseudo-progression and radionecrosis, from true disease progression [45,46]. However, their limited availability reduces their full integration into clinical practice. In addition to the aforementioned tracers, other radiopharmaceuticals used for prostate cancer evaluation, namely, [^11^C]/[^18^F]choline or [^18^F]fluorocyclobutane-1-carboxylic acid (FACBC or Fluciclovine), have been studied in patients with glioma [11]. PSMA ligands have also shown promising results in these patients, even when compared to other tracers.

In a pilot study by Verma et al. [19], 10 patients with suspected gliomas underwent both [^68^Ga]Ga-PSMA-11 and [^18^F]FDG PET/CT (1 to 5 days apart), revealing PSMA expression in all patients. Of these, seven glioblastoma (GBM, WHO grade IV) patients (*n* = 8 lesions) showed high-grade PSMA uptake (median TBR = 13.9), whereas the remaining three low-grade glioma (LGG, WHO grade II) lesions had low PSMA uptake (median TBR = 3.4). Moreover, all contrast-enhancing (CE) tumors on MRI showed high PSMA uptake, allowing clear visualization of the lesions on PET/CT images. When compared with [^18^F]FDG, all high-grade gliomas (HGGs) were FDG avid, whereas no LGGs were detected. Similarly, in a subsequent study, Liu et al. [20] evaluated 30 newly diagnosed and untreated patients with expanding intracranial lesions using [^68^Ga]Ga-PSMA-617 and [^18^F]FDG PET/CT. The sample included 14/30 LGGs and 16/30 HGGs. [^68^Ga]Ga-PSMA-617 mainly showed remarkable uptake in lesions or lesion margins in most of the 16 HGG patients (3/4 grade III; 11/12 grade IV). No relevant uptake was observed for the LGG patients (14/14 grade II). [^18^F]FDG uptake was able to detect 11/14 grade II, 4/4 grade III, and 11/12 grade IV lesions; however, in LGG patients, the uptake was mostly similar to that in the normal brain matter. All [^68^Ga]Ga-PSMA-617 and [^18^F]FDG parameters differed significantly between the LGG and HGG lesions, but PSMA SUV_max_ and SUV_mean_ were the most effective in differentiating HGGs from LGGs (PSMA area under curve—AUC—of 0.96 and 0.94 vs. 0.79 and 0.74 for FDG, respectively). Interestingly, [^68^Ga]Ga-PSMA-617 TBR_max_ in grade II lesions differed significantly between IDH wild type and mutant type (2.87 ± 0.77 vs. 0.8 ± 0.18), but no significant difference was observed between IDH mutant type and IDH mutant type with the co-deletion of 1p/19q (0.8 ± 0.18 vs. 0.89 ± 0.54). The difference between glioma grade and PSMA expression was also demonstrated in another study by Verma et al. [21] in thirteen treatment-naïve patients with suspected glioma and two recurrent gliomas. [^68^Ga]Ga-PSMA-11 PET/CT showed tracer uptake in all twelve HGGs and one of three LGGs with good TBR (range, 2.1–22.4; mean, 9.8), resulting in a significant difference in PSMA expression (SUV_max_ range in grade III and IV tumors was 2.65–30.2, mean 11.25; SUV_max_ range in grade I and II tumors was from non-avid to 2.9, mean 0.95). Conversely, most HGGs were [^18^F]FDG avid, whereas most of the LGGs were [^18^F]FDG non-avid or showed only low SUV_max_ values. Statistical analysis of the correlation between [^18^F]FDG uptake and PSMA expression with clinicopathological prognostic parameters showed a significant correlation between the SUV_max_ of both tracers with glioma grade, Ki-67 index, and IDH mutation status.

In our literature review, PSMA ligands seem to outperform [^18^F]FDG PET/CT, especially in the assessment of LGGs. However, although [^18^F]FDG PET/CT is still the most widely used radiopharmaceutical in clinical practice for PET imaging of cancer, there are several tracers based on neutral amino acid analogs (such as [^11^C]MET, [^18^F]FET, [^18^F]F-DOPA, and [^18^F]F-fluorothymidine ([^18^F]FLT)) that have shown improved diagnostic performance in the management of CNS tumors [46,47,48]. Therefore, comparative studies with amino acid radiotracers are of particular interest.

In a phase I/II pilot clinical trial (uncontrolled), Brighi et al. [22] compared the performance of [^68^Ga]Ga-PSMA-617 and [^18^F]FET PET/CT and contrast-enhanced magnetic resonance imaging (CE-MRI) in 10 patients with recurrent GBM. The mean biological tumor volume (BTV) delineated on the [^68^Ga]Ga-PSMA-617 images was slightly larger, although there was no significance (*p* = 0.063), than the respective mean BTV delineated on the [^18^F]FET PET images. This result was further confirmed by measuring the mean volumetric ratio between [^68^Ga]Ga-PSMA-617 BTV and [^18^F]FET BTV, which was 1.87 ± 1.1 (range: 0.3–4.3). The mean Dice similarity coefficient between [^68^Ga]Ga-PSMA-617 BTV and [^18^F]FET BTV was 0.6 ± 0.2 (range: 0.35–0.8), demonstrating the mismatch between the BTV margins delineated by the two tracers. The qualitative assessment revealed different uptake patterns for the two radiotracers, with [^68^Ga]Ga-PSMA-617 hotspots not always corresponding to [^18^F]FET, suggesting that neoangiogenesis is present in tumor regions that are not yet metabolically hyperactive. Furthermore, the [^68^Ga]Ga-PSMA-617 BTV was approximately four times larger than the CE tumor volume (*p* = 0.004), suggesting that [^68^Ga]Ga-PSMA-617 might accumulate in regions of early neoangiogenesis that are yet to progress to a stage where they present blood–brain barrier leakage. This suggests that PSMA could be a more useful imaging biomarker for secondary treatment planning and tumor delineation than [^18^F]FET and CE-MRI.

Finally, although the trial is still recruiting, [^68^Ga]Ga-PSMA-11 has also been compared with [^18^F]F-DOPA in the PAraDiGM trial to differentiate GBM recurrence from TRC, showing a good performance of [^68^Ga]Ga-PSMA-11 in the preliminary results in nine patients [49].

The significance of PSMA expression in the neovasculature of brain tumors remains to be fully determined; however, the pattern of expression suggests its functional role in the angiogenesis of glioma. In this scenario, if confirmed in future trials, PSMA-based PET/CT could potentially play a role in assessing response to anti-angiogenic therapy, such as bevacizumab, in delineating tumor regions most sensitive to external beam radiotherapy and in identifying glioma patients suitable for targeted [^177^Lu]Lu-PSMA therapy.

### 3.2. Breast Cancer

According to current oncological guidelines [50] and the growing body of clinical evidence, [^18^F]FDG PET/CT is, to date, the most commonly used radiopharmaceutical in breast cancer (BC) patients. Other radiopharmaceuticals, such as ^18^F-labeled estradiol ([^18^F]FES) or ^89^Zr-labeled Trastuzumab ([^64^Cu]Cu-DOTA-trastuzumab), have been proposed for the evaluation of the receptor status [51,52]. Recent research developments in nuclear medicine are driving efforts toward more specific imaging-based analysis, which implies more tailored treatment. Therefore, the possibility to target various metabolic and signaling pathways involved in the different diseases is becoming crucial in this regard. Data are also emerging on the use of PSMA-labeled PET/CT in BC, especially for patients who presented with negative receptor status. We report on the work by Medina-Ornelas et al. [23], published in 2020. The authors studied twenty-one patients with BC: four luminal A (LUM-A); four LUM-B HER2-positive; two LUM-B HER2-negative; and six HER2+; 5 triple-negative (TPN). [^68^Ga]Ga-PSMA-11 showed a lower detection rate (DR) than [^18^F]FDG PET/CT in all patients with LUM-A and LUM-B HER2(+/−). Conversely, both radiopharmaceuticals were able to detect all bone metastases and all lesions in TPN patients. The overall sensitivity and specificity were, respectively, 99.2% and 93.6% for [^18^F]FDG vs. 84% and 91.8% for [^68^Ga]Ga-PSMA-11, respectively, with similar accuracy ([^18^F]FDG AUC 0.86–0.95; [^68^Ga]Ga-PSMA-11 AUC 0.74–0.94). Another study by Arslan et al. [24] prospectively reported the results of forty-two TPN BC who underwent both [^68^Ga]Ga-PSMA-11 and [^18^F]FDG (thirty-six staging and six recurrent patients). A total of 46 of the 47 primary lesions were FDG avid (mean SUV_max_ 22.2 ± 15.2), whereas 38 of 47 showed PSMA expression (mean SUV_max_ 6.6 ± 3.4). Axillary lymph node (ALN) involvement was detected by both tracers in 57.4% (*n* = 27) of patients. However, patients with PSMA-positive primary tumors showed a higher rate of axillary lymph node (ALN) metastases compared to those with PSMA-negative primary tumors (64.7% vs. 25%, *p* = 0.035); distant organ metastases were observed in 24 patients (51.1%) with a mean SUV_max_ of 15.5 ± 11.6 for [^18^F]FDG versus 6.0 ± 2.9 for [^68^Ga]Ga-PSMA-11. There was also a negative correlation between the [^68^Ga]Ga-PSMA-11 mean SUV_max_ of the primary lesion and the Ki-67 index (r = −0.450). The overall sensitivity of [^18^F]FDG was 97.6% compared to 92.7% for [^68^Ga]Ga-PSMA-11.

Recently, [^18^F]PSMA-1007 was compared to [^18^F]FDG PET/CT in 10 TNP patients by Andryszak et al. [25]. In the only treatment-naïve patient, both radiopharmaceuticals showed a pathological uptake in the right breast tumor (SUVmax 6.0 and TBR 2.8 for [^18^F]PSMA-1007; SUVmax 3.4 and TBR 2.8 for [^18^F]FDG), but only [^18^F]PSMA-1007 showed an additional pathological uptake in the same breast not detected by [^18^F]FDG PET/CT. All patients treated with chemotherapy (4/10) showed local uptake of both tracers in the breast. Specifically, in three of them, the uptake of [^18^F]FDG was higher than that of [^18^F]PSMA-1007, whereas, in the fourth patient, the radiopharmaceutical accumulation was similar. The number of patients with PET-positive lymph nodes was the same in both modalities (5/8 patients); however, in one patient, [^18^F]PSMA-1007 PET/CT detected one extra positive lymph node than [^18^F]FDG, while in two other cases, [^18^F]FDG showed more lesions than [^18^F]PSMA-1007 PET/CT. Considering distant metastases, both radiopharmaceuticals were able to identify larger metastatic lesions (>5 mm); however, in patients with bone metastases, SUV_max_ and TBR values were higher in [^18^F]PSMA-1007 than [^18^F]FDG, and liver metastases (up to 67 mm) showed higher [^18^F]PSMA-1007 uptake than [^18^F]FDG (SUV_max_ = 17.3 vs. 4.1; TBR = 3.4 vs. 1.8). Furthermore, ten small brain metastases (diameter ranging from 4 to 7 mm) were only detected through [^18^F]PSMA-1007 PET/CT ([^18^F]FDG-negative), showing a clear added value of [^18^F]PSMA-1007 PET/CT over [^18^F]FDG PET/CT in the assessment of brain metastases.

According to the reviewed literature, PSMA PET/CT may, therefore, play a role in the future evaluation of patients with HR-negative tumor phenotypes, such as BC TPN, both for staging and monitoring treatment response to selective angiogenesis inhibition.

### 3.3. Thyroid Cancer

[^18^F]FDG PET/CT is recommended by the American Thyroid Association guidelines for the detection of tumor recurrence and metastases in radioactive iodine refractory differentiated thyroid cancer (DTC) [53]. However, its sensitivity varies from 68.8 to 82%, and a false-negative rate of 8–21.1% has been reported in patients with “thyroglobulin elevated negative iodine scintigraphy” (TENIS) syndrome [54,55,56]. Recently, immunohistochemistry studies have shown that PSMA is expressed in the microvasculature of thyroid cancer (TC), while it is not expressed in normal thyroid tissue and benign thyroid tumors [57]. Preliminary studies have shown heterogeneous PSMA uptake in TC patients, often depending on the histologic type. Verma et al. [26] studied 10 metastatic DTC patients who underwent prospective evaluation with radioiodine ([^131^I]I) scintigraphy, [^18^F]FDG, and [^68^Ga]Ga-PSMA-11 PET/CT. Lesions were iodine avid in eight patients, while in two patients were classified as TENIS. All patients with iodine-avid metastatic disease showed substantial PSMA uptake. [^68^Ga]Ga-PSMA-11 PET/CT detected 30/32 (93.75%) total lesions (SUV_max_ ranging from 4.9 to 101.8—median SUV_max_ 31.35), whereas [^18^F]FDG PET/CT was positive in 23/32 lesions (81.85%). Conversely, in another recent head-to-head comparison between [^68^Ga]Ga-PSMA-11 and [^18^F]FDG PET/CT published by Shi et al. [27], the authors prospectively enrolled 23 DTC and 17 radioiodine refractory DTC (RAIR-DTC) subjects for a total of 72 lesions. The detection rate of both DTC and RAIR-DTC was lower for [^68^Ga]Ga-PSMA-11 than [^18^F]FDG (60% vs. 90%, *p* = 0.004; 59.4% vs. 96.9%, *p* < 0.001, respectively). In addition, immunohistochemistry showed a significantly higher PSMA expression in RAIR-DTC than in DTC, but no significant correlation was observed between PSMA expression and SUV_max_. In another study, Pitalua-Cortes et al. [28] performed a retrospective comparison between post-therapeutic [^131^I]I whole-body scan (with complementary single photon emission tomography—SPECT/CT) and [^68^Ga]Ga-PSMA–11 PET/CT in 10 metastatic DTC. Sixty-four metastatic lesions were analyzed: 67.2% had papillary histology and 32.8% were the follicular type. The most affected site of metastases was bone (57.8%), followed by lung (17.2%), lymph nodes (7.8%), postoperative thyroid bed (4.7%), brain (4.7%), and others (7.8%). [^68^Ga]Ga-PSMA–11 PET/CT detected 64/64 lesions, all of them were also identified by CT, whereas [^131^I]I SPECT/CT detected 55/64 lesions. Discordant findings were described for lesions localized in the lung (44.4%), brain (22.2%), postoperative thyroid bed (11.1%), lymph nodes (11.1%), and bone (11.1%). Another prospective comparison between [^131^I]I scintigraphy and both [^18^F]FDG and [^68^Ga]Ga-PSMA-11 PET/CT or PET/MRI was performed by Lawhn-Heath and colleagues [29] on 11 adults with a history of pathology-proven TC showing abnormal uptake on [^18^F]FDG PET and/or [^131^I]I scintigraphy performed in the 12 months before. Seven out of eleven patients had differentiated disease (three papillary TC, two follicular TC, and two Hurthle cell) and 4/11 had de-differentiated disease (two poorly differentiated papillary and two anaplastic). Out of 43 lesions, 41 were [^18^F]FDG-positive (detection rate 95.3%) and 28 were [^68^Ga]Ga-PSMA-11-positive (detection rate 65.1%). In a recent prospective study by Feng et al. [30], the authors conducted both in vitro and in vivo studies to correlate TC subtypes with PSMA expression (by immunohistochemical staining) and uptake (in terms of SUV_max_) levels at PET/CT. The study showed no significant differences in [^68^Ga]Ga-PSMA-11 PET/CT uptake between patients with de-differentiated thyroid cancer and those with well-differentiated thyroid cancer, although PSMA immunohistochemical staining results showed significant differences in PSMA expression between well-differentiated and de-differentiated thyroid cancer. The [^18^F]FDG SUV_max_ was significantly higher in PSMA-positive lesions compared to PSMA-negative ones (8.08 ± 7.74 vs. 5.67 ± 4.23, *p* = 0.01). In addition, [^68^Ga]Ga-PSMA-11-positive patients had higher Tg levels (307.1 ± 183.4 vs. 118.0 ± 116.1, *p* = 0.002).

The evidence in the literature so far is inconclusive, and further studies are needed to evaluate the possible role of radiolabeled PSMA PET/CT in thyroid cancer, particularly in TENIS or anaplastic TC patients.

### 3.4. Adenoid Cystic Carcinoma

[^18^F]FDG plays a major role in staging and response assessment of salivary gland carcinoma; however, it is generally more accurate in evaluating salivary duct carcinoma (SDC) compared to adenoid cystic carcinoma (AdCC), which has a lower [^18^F]FDG uptake [58].

PSMA expression was found to be associated with tumor neovasculature in both AdCC and SDC [59]. The only available study comparing [^68^Ga]Ga-PSMA-11 and [^18^F]FDG PET/CT was performed by Shamim et al. [31] in 17 AdCC patients. Among the 14/17 patients who underwent both radiotracers, the mean SUV_max_ of the primary tumor was higher in [^18^F]FDG compared to [^68^Ga]Ga-PSMA-11 PET/CT (5.6 ± 4.7 versus 4.8 ± 3.2). Interestingly, while lung (*n* = 7) and lymph nodes (*n* = 5) metastases were detected in both PET scans, cerebellar, meningeal, and bone metastases were only detected in [^68^Ga]Ga-PSMA-11 PET/CT.

### 3.5. Hepatocellular Carcinoma

To date, the diagnosis of hepatocellular carcinoma (HCC) is mainly based on functional images, such as ultrasound, CT, and MRI. However, since morphological characteristics do not incorporate the biological behavior of the tumor, PET imaging agents have been evaluated as well [60]. Generally, [^18^F]FDG PET is not considered useful for HCC detection and it is not currently recommended for the diagnosis of HCC because of the high rate of false-negative results; conversely, [^11^C]/[^18^F]choline PET has demonstrated a higher detection rate and sensitivity compared to [^18^F]FDG [61,62]. In this scenario, radiolabeled PSMA PET/CT has recently shown promising results in comparison with [^18^F]FDG. In 2021, Gündoğan et al. [32] studied 14 HCC patients with MRI, [^18^F]FDG, and [^68^Ga]Ga-PSMA-11 PET/CT. By a visual analysis, [^68^Ga]Ga-PSMA-11 was able to detect more primary and metastatic lesions than [^18^F]FDG PET/CT. In fact, the number of liver lesions in [^68^Ga]Ga-PSMA-11 PET/CT and MRI were significantly higher than [^18^F]FDG PET/CT (*p* = 0.042 and 0.026, respectively), but there was no statistically significant difference between the number of liver lesions in MRI and [^68^Ga]Ga-PSMA-11 PET/CT. Moreover, tumor to abdominal aorta (T/A) and tumor to the gluteal muscle (T/G) SUV_max_ ratio were significantly higher in [^68^Ga]Ga-PSMA-11 compared to [^18^F]FDG (*p* = 0.002 and 0.002, respectively). These results support the potential use of [^68^Ga]Ga-PSMA-11 PET/CT as a complementary tool to MRI in HCC T staging, particularly in the evaluation of multicentric tumors. In contrast, in a cohort of 19 HCC patients studied by Kuyumcu et al. [33], no significant differences between [^18^F]FDG and [^68^Ga]Ga-PSMA-11 were demonstrated. On a per-patient analysis, the two tracers only differed for one patient (15 [^18^F]FDG-positive patients vs. 16 [^68^Ga]Ga-PSMA-positive patients) and the only extrahepatic lesion (a metastatic lymph node) was detected by both tracers. Considering visual and quantitative evaluation, nine patients showed higher PSMA uptake compared to [^18^F]FDG, and the latter was higher than [^68^Ga]Ga-PSMA-11 in the other four patients. A comparison of mean SUV_max_ and TBR values revealed no statistically significant difference (*p* > 0.1), and both tracers’ uptake correlated with prognosis.

Thus, according to the literature, PSMA expression in advanced HCC can be detected by [^68^Ga]Ga-PSMA-11 PET/CT, but it does not have clear superiority over [^18^F]FDG PET/CT. Nevertheless, its potential utility as a complementary tool to MRI for the T-staging of HCC should be further investigated.

### 3.6. Colorectal, Gastric, and Pancreatic Cancer

Colorectal cancer (CRC) staging is usually carried out by contrast-enhanced CT (ceCT). Additional abdominopelvic ultrasonography or MRI is the preferred choice in case of liver metastases to accurately define their number and localization. The same radiological techniques are used for response assessment after therapy [63]. As stated in the current guidelines, [^18^F]FDG PET/CT can be useful in patients with increased tumor markers without evidence of metastatic disease or to define the extent of metastatic disease on potentially resectable metastases [64]. For gastric and esocrine pancreatic cancers, [^18^F]FDG PET/CT does not represent the modality of choice for first diagnosis and further patient management [65,66].

Exploring alternative radiopharmaceuticals, such as PSMA-based tracers, could open the way to new effective diagnostic strategies, although preliminary comparison studies so far have not demonstrated a clear superior detection rate of PSMA over [^18^F]FDG. In a study by Cuda et al. [34], ten CRC patients were included, eight of whom had detectable lesions in [^18^F]FDG PET/CT or ceCT but missed in [^68^Ga]Ga-PSMA-11 PET/CT. In only one patient, bone metastasis showed significantly higher avidity on [^68^Ga]Ga-PSMA-11 PET than other soft-tissue and visceral lesions. However, bone lesion avidity was still significantly lower than [^18^F]FDG PET/CT. Similar conclusions were drawn by a recent prospective study by Vuijk et al. [35], who evaluated the feasibility of [^18^F]DCFPyL PET/CT imaging to detect gastrointestinal cancers, including colon, pancreatic, and gastric cancer. As part of this trial, ten patients underwent both [^18^F]DCFPyL and [^18^F]FDG PET/CT before surgery (four colon and three pancreatic cancer patients) or before neoadjuvant therapy (three gastric cancer patients). According to tumor differentiation, two patients had a well-differentiated adenocarcinoma, three were scored as well/moderate, two as moderate, and three as poor-differentiated. Results from this study demonstrated that [^18^F]DCFPyL PET/CT was able to detect the primary tumor in seven out of ten patients (3/4 colon, 1/3 gastric, 3/3 pancreatic cancers), whereas [^18^F]FDG PET/CT was positive in six out of nine patients (3/4 colon, 1/2 gastric, 2/3 pancreatic tumors). However, due to the low contrast and high level of uptake in the surrounding tissue, the visual distinction of these tumors from the background was difficult, and the SUV_max_ and TBR in [^18^F]DCFPyL PET/CT were significantly lower compared to [^18^F]FDG PET/CT. In addition, no correlation between PSMA expression in the tumor specimen and SUV_max_ in [^18^F]DCFPyL PET/CT was found. Conversely, Krishnaraju et al. [36] assessed whether [^68^Ga]Ga-PSMA-11 PET can be used as a new non-invasive diagnostic tool to differentiate between malignant and benign pancreatic lesions. A total of 40 patients prospectively underwent whole-body [^18^F]FDG PET/CT and regional [^68^Ga]Ga-PSMA-11 PET/CT. By a visual analysis, [^18^F]FDG PET/CT was positive in 26 lesions (65%), while [^68^Ga]Ga-PSMA-11 PET/CT was positive in 20 (50%) lesions. Findings were concordant in 30/40 (75%) lesions (eighteen positives and twelve negatives) and discordant in the remaining 10/40 (25%) lesions (solely [^18^F]FDG positivity in eight lesions, solely [^68^Ga]Ga-PSMA-11 positivity in two lesions). Overall, 19 findings were malignant and 21 were benign in the histopathology/cytopathology examination, showing a better diagnostic performance of [^68^Ga]Ga-PSMA-11 over [^18^F]FDG PET/CT for characterizing pancreatic lesions. The overall sensitivity of both [^68^Ga]Ga-PSMA-11 PET/CT and [^18^F]FDG PET/CT was high (94.7% vs. 89.5%), while the specificity was higher for [^68^Ga]Ga-PSMA-11 (90% vs. 57.1%) for the detection of primary pancreatic neoplasm.

Generally, the available data on the role of radiolabeled PSMA in gastrointestinal cancers are inconclusive. However, more studies considering homogenous samples (i.e., pancreatic tumors) should be performed.

### 3.7. Renal Cell and Urothelial Cancer

Renal cell cancer (RCC) is a highly vascular tumor in which ceCT remains the modality of choice for staging, assessment of treatment response, and recurrence evaluation [67]. Although international guidelines do not recommend PET imaging in RCC, the available evidence supports [^18^F]FDG or radiolabeled PSMA PET use for disease characterization. According to recent studies, RCC neovasculature seems to express PSMA, representing a potential diagnostic target for radiolabeled PSMA PET/CT [68]. In a paper by Aggarwal et al. [37], 37 biopsy-proven RCC patients with known or suspected distant metastases who underwent [^68^Ga]Ga-PSMA-11, ceCT, and [^18^F]FDG PET/CT for staging/restaging were prospectively recruited. [^68^Ga]Ga-PSMA-11 PET identified more lesions than ceCT in 10/37 (+27%) patients and was able to detect a significantly higher number of bone lesions (100% vs. 45%, *p* < 0.001), whereas ceCT detected a higher number of liver lesions (20.8% vs. 100%, *p* < 0.001). When compared to [^18^F]FDG PET, [^68^Ga]Ga-PSMA-11 detected more lesions (312 vs. 202, *p* < 0.001), with 113 PSMA-positive/FDG-negative lesions and only 14 PSMA-negative/FDG-positive lesions. Furthermore, significantly higher SUV_max_ (6.9 vs. 5.2, *p* < 0.001), SUV_peak_ (4.4 vs. 3.8, *p* = 0.004), and TBR (5.7 vs. 3.8, *p* < 0.001) were registered in [^68^Ga]Ga-PSMA-11 compared to [^18^F]FDG PET, even with a prognostic value. In another study with a smaller sample size (11 patients), PET imaging was confirmed to be more accurate than conventional imaging for tumor burden characterization, with fewer false-positive findings. When PET tracers were compared, concordant [^18^F]FDG and [^68^Ga]Ga-PSMA-11 uptake in metastatic RCC lesions were observed in 9/11, with the remaining two cases showed discordant uptake favoring PSMA [38]. In a retrospective study by Liu et al. [39], a total of 15 patients with suspected local recurrence of clear-cell RCC or metastases after surgery underwent both [^18^F]F-DCFPyL and [^18^F]FDG PET/CT. [^18^F]F-DCFPyL PET/CT was statistically more accurate (*p* = 0.002) in the detection of bone lesions. The average SUV_max_ and TBR of pathological foci in [^18^F]F-DCFPyL were significantly higher than in [^18^F]FDG for soft tissue lesions (SUV_max_
*p* = 0.005; TBR *p* = 0.028) and bone lesions (SUVmax *p* = 0.001; TBR *p* = 0.001). With regards to the impact on clinical management, a study by Udovicich et al. [40] showed that overall, 30 patients (49%) had a change in management due to PSMA PET/CT. The most common change was from an initial plan for metastasis-directed therapy (MDT; stereotactic ablative radiotherapy—SABR—or metastasectomy), systemic therapy, or surveillance (15 patients). Conversely, nine patients, who were candidates for systemic therapy or surveillance before PSMA PET/CT, were shifted to MDT instead, according to the PET/CT results. Furthermore, four patients received SABR at additional sites and two patients received SABR at fewer sites. In the same study, [^68^Ga]Ga-PSMA-11 PET/CT detected more lesions than CT in 15 patients (25%) and fewer lesions in 16 patients (26%). A subgroup of 40 patients underwent both [^18^F]FDG and [^68^Ga]Ga-PSMA-11 PET with a detection rate of 75% (30/40 patients) and 88% (35/40 patients; *p* = 0.18), respectively. Twenty-eight patients had PSMA- and FDG-positive disease and three patients had PSMA- and [^18^F]FDG-negative disease. Seven patients had discordant PSMA-positive/[^18^F]FDG-negative disease and two patients had discordant PSMA-negative/[^18^F]FDG-positive disease. SUV characteristics were compared for the 28 patients with PSMA- and [^18^F]FDG-positive disease, with higher SUV_max_ values for PSMA PET/CT (15.2 vs. 8.0; *p* = 0.02). In this scenario, it is interesting to mention a prospective study including 72 clear-cell RCC patients who underwent both [^18^F]FDG and [^68^Ga]Ga-PSMA-11 PET/CT [41]. In these patients, the SUV_max_ of lesions at [^68^Ga]Ga-PSMA-11 PET/CT significantly predicted subgroups of tumor necrosis, sarcomatoid or rhabdoid features, and adverse pathology (all *p* < 0.01) with an AUC of 0.85 (cutoff value = 25.3, *p* < 0.001; Delong test z = 2.709, *p* = 0.007) for tumor necrosis and an AUC of 0.90 (cutoff value = 25.26, *p* < 0.001; Delong test z = 3.433, *p* < 0.001) for adverse pathology. Wang et al. [42] demonstrated that for primary clear-cell RCC patients, [^68^Ga]Ga-P16-093 (a small-molecule PSMA ligand) had a significantly higher detection rate (19/22 vs. 13/22, *p* = 0.031) and higher tumor uptake (15.7 ± 9.0 vs. 5.1 ± 3.4, *p* < 0.001) than [^18^F]FDG PET/CT. In addition, the SUV_max_ of the primary tumor in [^68^Ga]Ga-P16-093 and [^18^F]FDG PET/CT significantly correlated with the pT stage (r = 0.55, *p* = 0.008; r = 0.51, *p* = 0.014, respectively) and WHO/ISUP grade (for [^68^Ga]Ga-P16-093, r = 0.57, *p* = 0.006; for [^18^F]FDG, r = 0.49, *p* = 0.02). For metastatic clear-cell RCC patients, [^68^Ga]Ga-P16-093 PET/CT also demonstrated a better detection rate (21/22 vs. 14/22, *p* = 0.008) and higher tumor uptake (11.0 ± 6.4 vs. 4.4 ± 2.7, *p* < 0.001) than [^18^F]FDG PET/CT. Also, the SUV_max_ in [^68^Ga]Ga-P16-093 PET/CT had a significant association with PSMA expression (by immunohistochemistry) in primary clear-cell RCC (r = 0.8, *p* < 0.001) and metastatic clear-cell RCC (r = 0.62, *p* = 0.03).

On the contrary, in a pilot study evaluating urothelial carcinoma of the upper tract (UTUC, a relatively rare form of urothelial carcinoma), [^18^F]FDG PET/CT demonstrated a more effective detection of UTUC foci compared to [^68^Ga]Ga-PSMA-11 PET/CT, with a higher SUV_max_ (respectively, SUV_max_ of 18.5 ± 6.7 vs. 4.4 ± 1.45, *p* < 0.01). Immunohistochemical analysis revealed a statistically significant difference in the expression of PSMA and GLUT1 in UTUC (*p* = 0.048), with higher pathological grades showing more intense GLUT1 staining than PSMA (75% vs. 12.5%) [43].

Considering the therapeutic implications, PSMA PET could open a new scenario for RCC patients thanks to its higher detection rate compared to conventional imaging and [^18^F]FDG, as shown in this review. In other words, the possibility of using a more sophisticated tool to detect disease foci could contribute to better management of RCC patients. The data are promising, and studies are warranted.

## 4. Perspectives and Conclusions

PSMA PET/CT imaging has a well-established role as an up-and-coming target for molecular imaging in PCa. It represents, nowadays, the gold standard for staging high-risk patients [69], biochemical recurrence [70,71], and radioligand therapy selection [72]. There is also preliminary evidence regarding the potential of PSMA PET in PCa biopsy guidance [73,74] and response assessment in several therapeutic settings [75,76].

There is increasing awareness that this glycoprotein appears to be overexpressed in different tumors, albeit the name “PSMA” suggests a specificity for PCa. Particularly, using immunohistochemistry and monoclonal antibodies targeting PSMA (both extracellular and intracellular domains), this protein’s upregulation has been demonstrated on endothelial cells of the neovasculature of a variety of solid tumors, while no endothelial expression is described in physiological conditions [77,78]. It has been shown that it may facilitate endothelial cell sprouting and invasion by regulating lytic proteases able to cleave the extracellular matrix [79]. Thus, there is a growing interest in PSMA PET oncological applications beyond PCa, as it could offer solutions to some open diagnostic challenges.

Indeed, the main evidence was observed in RCC. In this setting, the possibility of overcoming the known limitations of [^18^F]FDG PET/CT could open new scenarios for diagnosis and therapy. Most of the available data suggest a higher detection rate of PSMA PET compared to [^18^F]FDG [37,39,42], leading to a change in the management of RCC patients, as shown in the study by Udovicich et al. [40]. PSMA PET may also improve the assessment of other tumors, such as gliomas, especially in LGG patients [19,20]. Therefore, PSMA may also play a role in the early assessment of new areas of early neoangiogenesis in patients showing progression of disease [22]. In this context, the possible application of PSMA in the detection of brain metastases should also be considered [23,28,31]. Further data are needed to assess the potential role of PSMA PET in BC patients, which may provide a novel therapeutic vascular target in patients with negative receptor status [23,24,25]. Unclear applications of PSMA-PET include TC, where it may be useful in selected patients with either [^131^I]I-negative or [^18^F]FDG-negative tumor lesions [26,28]. Finally, even if PSMA-based radiopharmaceuticals show a potential role in HCC patients [32,33], there is no solid evidence for other gastrointestinal tumors [34,35].

The present review expresses an effort to collect major data on PSMA “non-exclusivity” to PCa, opening a window on a new horizon on the potential use of PSMA-PET/CT beyond prostate tumors. Future perspectives include not only diagnostic issues on molecular characterization of tissues, staging, and restaging but also therapeutic implications. The occurring overexpression of this glycoprotein in the neovasculature of multiple malignancies could contribute to the prospective estimation of therapy response and therapy monitoring of anti-angiogenic drugs, too. Still, its possible role as a biomarker of tumor neo-angiogenesis is under debate and assessment. In recent years, PSMA-targeted radioligand therapies, namely, [^177^Lu]Lu-PSMA-617, have gained an established role in PCa treatment, and the exploration of PSMA radioligand therapy for other tumors is limited to date [18,80]. High and consistent PSMA uptake at PET/CT is the prerequisite to recruiting PCa patients for this treatment; therefore, it is possible to hypothesize such an application beyond PCa. This could potentially have a role in patients, such as GBM or RAIR-TC patients, where the lack of effective treatments could make them suitable candidates for radioligand therapy. However, the therapeutic effect in PCa is mainly achieved thanks to PSMA overexpression in the PCa cell membranes, and in other cancers it is prevalently located in the tumor-associated endothelium. Moreover, a durable bond of the radionuclide to the target lesions critically influences the effectiveness of radioligand treatment, but no data are available on the kinetics and binding affinity of beta-emitting PSMA radioligands to the endothelium. With these regards, dosimetry studies could offer a proper answer in this open field of potential PSMA radioligand therapy beyond PCa.

In conclusion, the future of molecular imaging and therapy of tumors other than PCa is also bright for PSMA ligands, but prospective clinical trials with larger sample sizes are essential to further investigate their potential and the following possible clinical applications.

## Figures and Tables

**Figure 1 diagnostics-14-01002-f001:**
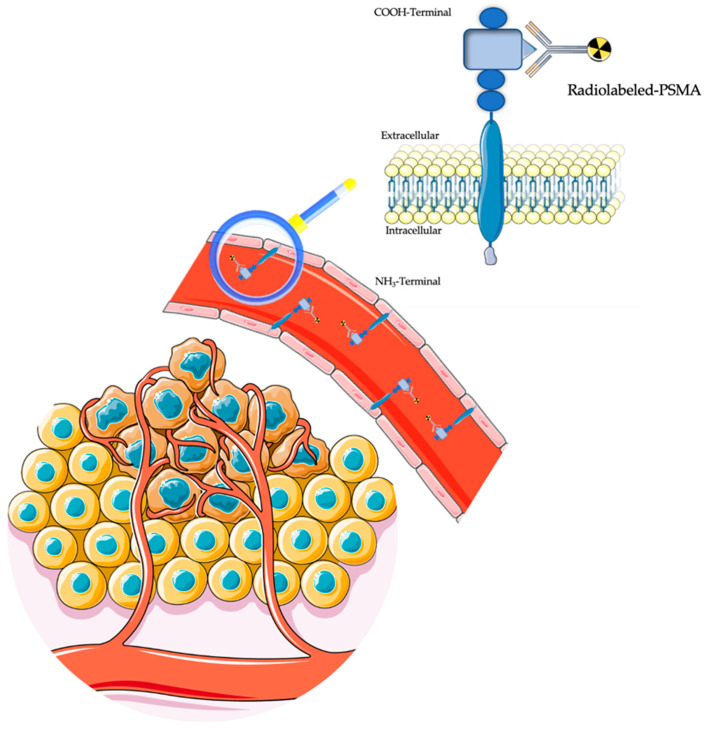
Schematic representation of PSMA expression from non-prostatic tumor-associated neovasculature. Selected elements in this figure were adapted from pictures provided by Servier Medical Art; Servier; https://smart.servier.com/ (accessed on 15 February 2024), licensed under a Creative Commons Attribution 4.0 Unported License.

**Figure 2 diagnostics-14-01002-f002:**
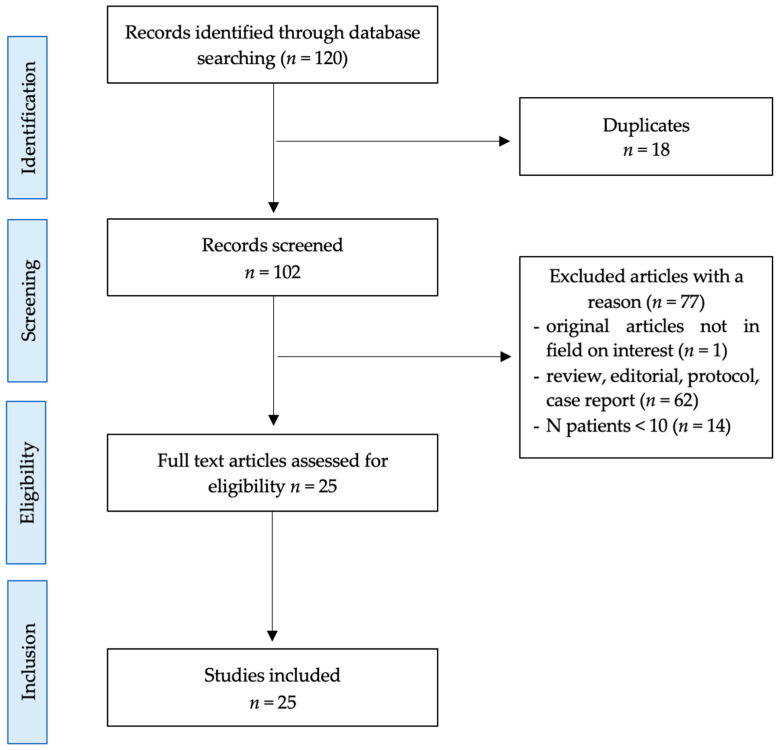
Flowchart of research strategy and study selection.

**Table 1 diagnostics-14-01002-t001:** The main characteristics of the included studies classified according to primary tumor.

Authors	Tumor Type	N°	PSMA-Ligand	Comparison	Detection Rate (PSMA vs. Comparison)	Main Findings
Verma et al. [19]	Glioma	10	[^68^Ga]Ga-PSMA-11	[^18^F]FDG	100% vs. 70%	[^68^Ga]Ga-PSMA-11 was able to better detect LGGs compared to [^18^F]FDG
Liu et al. [20]	Glioma	30	[^68^Ga]Ga-PSMA-617	[^18^F]FDG	93% vs. 86%	[^68^Ga]Ga-PSMA-617 SUV_max_ and SUV_mean_ were the most effective for differentiating HGGs from LGGs ([^68^Ga]Ga-PSMA-617 AUC of 0.96 and 0.94; [^18^F]FDG AUC of 0.79 and 0.74, respectively)
Verma et al. [21]	Glioma	15	[^68^Ga]Ga-PSMA-11	[^18^F]FDG	87% vs. 80%	Correlation between [^68^Ga]Ga-PSMA-11 and [^18^F]FDG SUV_max_ with glioma grade, Ki-67 index, and IDH mutation status
Brighi et al. [22]	Glioma	10	[^68^Ga]Ga-PSMA-617	[^18^F]FET	/	[^68^Ga]Ga-PSMA-617 BTV covers the CE tumor volume and extends to adjacent regions of non-enhancing tumor resulting in a BTV approximately four times larger than the CE tumor volume (*p* = 0.0039)
Medina-Ornelas et al. [23]	Breast Cancer	21	[^68^Ga]Ga-PSMA-11	[^18^F]FDG	70% vs. 100%	The overall sensitivities and specificities were, respectively, 99.2% and 93.6% for [^18^F]FDG vs. 84% and 91.8% for [^68^Ga]Ga-PSMA-11
Arslan et al. [24]	Breast Cancer	42	[^68^Ga]Ga-PSMA-11	[^18^F]FDG	81% vs. 98% (primary)	The overall sensitivity was 97.6% for [^18^F]FDG PET/CT vs. 92.7% for [^68^Ga]Ga-PSMA-11
Andryszak et al. [25]	Breast Cancer	10	[^18^F]PSMA-1007	[^18^F]FDG	/	The study showed a comparable uptake of [^18^F]PSMA-1007 and [^18^F]FDG in primary and metastatic lesions
Verma et al. [26]	Thyroid Cancer	10	[^68^Ga]Ga-PSMA-11	[^18^F]FDG	93.75% vs. 81.85%	All patients with iodine-avid metastatic disease showed substantial PSMA uptake 70% of lesions that showed PSMA expression was localized to the bones
Shi et al. [27]	Thyroid Cancer	40	[^68^Ga]Ga-PSMA-11	[^18^F]FDG	DTC 60% vs. 90% RAIR-DTC 59.4% vs. 96.9%	[^68^Ga]Ga-PSMA-11 showed a lower detection rate than [^18^F]FDG PET/CT There was a difference in PSMA expression levels between DTC and RAIR-DTC, but the difference was not reflected on [^68^Ga]Ga-PSMA-11 PET/CT
Pitalua-Cortes et al. [28]	Thyroid Cancer	10	[^68^Ga]Ga-PSMA-11	^131^I WBS + SPECT/CT	100% vs. 86%	[^68^Ga]Ga-PSMA-11 was superior in identifying metastatic lesions compared to ^131^I SPECT/CT
Lawhn-Heath et al. [29]	Thyroid Cancer	11	[^68^Ga]Ga-PSMA-11	[^18^F]FDG	65.1% vs. 95.3%	Thyroid cancer subtypes did not predict PSMA uptake, and radiotracer uptake differed between patients and lesions
Feng et al. [30]	Thyroid Cancer	42	[^68^Ga]Ga-PSMA-11	[^18^F]FDG	/	[^68^Ga]Ga-PSMA-11 uptake correlated with higher [^18^F]FDG SUV_max_ and Tg levels
Shamim et al. [31]	Adenoid Cystic Carcinoma	17	[^68^Ga]Ga-PSMA-11	[^18^F]FDG	94% vs. 93% (primary)	Cerebellar, meningeal metastasis, and bone lesions were detected only on [^68^Ga]Ga-PSMA-11 but were not visualized on [^18^F]FDG
Gündoğan et al. [32]	Hepatocellular Carcinoma	11	[^68^Ga]Ga-PSMA-11	[^18^F]FDG	/	[^68^Ga]Ga-PSMA-11 is superior to [^18^F]FDG PET/CT in the staging of HCC
Kuyumcu et al. [33]	Hepatocellular Carcinoma	19	[^68^Ga]Ga-PSMA-11	[^18^F]FDG	84% vs. 79%	PSMA expression in advanced HCC can be demonstrated by [^68^Ga]Ga-PSMA-11 PET but is not superior to [^18^F]FDG
Cuda et al. [34]	Colorectal Carcinoma	10	[^68^Ga]Ga-PSMA-11	[^18^F]FDG	/	The study suggested a low PSMA avidity due to consistently low PSMA expression in CRC tumors
Vuijk et al. [35]	Gastrointestinal Tumors *	11	[^18^F]DCFPyL	[^18^F]FDG	60% vs. 100%	[^18^F]FDG PET/CT was superior in detecting colon, gastric, and pancreatic cancers
Krishnaraju et al. [36]	Pancreas	40	[^68^Ga]Ga-PSMA-11	[^18^F]FDG	94% vs. 88% (tumor)	The overall sensitivity of both [^68^Ga]Ga-PSMA-11 and [^18^F]FDG PET/CT was high (94.7% vs. 89.5%), while the specificity was higher for [^68^Ga]Ga-PSMA-11 compared with [^18^F]FDG (90% vs. 57.1%) PET/CT for the detection of primary pancreatic neoplasm
Aggarwal et al. [37]	Renal Cell Carcinoma	37	[^68^Ga]Ga-PSMA-11	[^18^F]FDG	312 vs. 202	[^68^Ga]Ga-PSMA-11 PET/CT showed significantly higher SUV_max_ than [^18^F]FDG
Tariq et al. [38]	Renal Cell Carcinoma	11	[^68^Ga]Ga-PSMA-11 or [^18^F]PSMA-1007	[^18^F]FDG	60% vs. 80% (primary) 100% vs. 82% (metastases)	[^68^Ga]Ga-PSMA-11 and [^18^F]FDG PET/CT were mostly concordant for assessment of primary and metastatic RCC. PET imaging led to a change in patients’management.
Liu et al. [39]	Renal Cell Carcinoma	15	[^18^F]DCFPyL	[^18^F]FDG	100% vs. 61% (soft tissue and bone mets)	[^18^F]DCFPyL was statistically better (*p* = 0.002) at detecting bone lesions SUV_max_ and TBR were significantly higher than that of [^18^F]FDG for soft tissue lesions and bone lesions (*p* = 0.001)
Udovicich et al. [40]	Renal Cell Carcinoma	40	[^68^Ga]Ga-PSMA-11 or [^18^F]DCFPyL	[^18^F]FDG	88% vs. 75%	PSMA PET/CT detected additional metastases compared to CT in 25% of patients and registered a significantly higher SUV_max_ than [^18^F]FDG
Chen et al. [41]	Renal Cell Carcinoma	62	[^68^Ga]Ga-PSMA-11	[^18^F]FDG	/	The SUV_max_ of [^68^Ga]Ga-PSMA-11 PET/CT was more effective than [^18^F]FDG PET/CT in identifying tumor necrosis and adverse pathology
Wang et al. [42]	Renal Cell Carcinoma	42	[^68^Ga]Ga-P16-093	[^18^F]FDG	86% vs. 59% (primary) 95% vs. 64% (metastic)	[^68^Ga]Ga-P16-093 PET/CT demonstrated a significantly higher detection rate and higher tumor uptake either in primary and metastatic RCC
Lin et al. [43]	Urothelial Carcinoma	25	[^68^Ga]Ga-PSMA-11	[^68^Ga]Ga-LNC1007; [^18^F]FDG	LNC1007 vs. FDG: 13/17 vs. 4/17, *p* = 0.005; LNC1007 vs. PSMA: 9/11 vs. 6/11, *p* = 0.361	In the [^68^Ga]Ga-LNC1007 and [^68^Ga]Ga-PSMA-11 PET/CT group, the detection rates were 77.8% by patient-based analysis and 81.8% by lesion-based analysis for LNC1007 and 66.7% by patient-based analysis and 54.5% by lesion-based analysis for PSMA

Legend: AUC, area under the curve; BTV, biological tumor volume; CRC, colorectal cancer; CT, computed tomography; CE, contrast enhanced; DTC, differentiated thyroid cancer; FDG, fluorodeoxyglucose; HCC, hepatocellular carcinoma; HGG, high-grade glioma; LGG, low-grade glioma; PET, positron emission tomography; PSMA, prostate-specific membrane antigen; RAIR-TC, radioiodine refractory thyroid cancer; RCC, renal cancer cell; SPECT, single photon emission tomography; SUV, standardized uptake value. * Colorectal, gastric, and pancreatic cancers.

## Data Availability

Not applicable.

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
