# Peer review of "Prostate-Specific Membrane Antigen Positron Emission Tomography Oncological Applications beyond Prostate Cancer in Comparison to Other Radiopharmaceuticals"

_diagnostics, 2024, doi:10.3390/diagnostics14101002_

Round 1

Reviewer 1 Report

Comments and Suggestions for Authors

The authors provide a comprehensive systematic review of PSMA PET oncological applications beyond prostate cancer. The review is meticulously written, and the paper is clear and concise. Moreover, the topic is interesting and relevant. I recommend the publication of the paper after a minor revision.

- Table 1: Please provide a separate legend for opening the acronyms of the primary tumors. Moreover, it would improve clarity if reference numbers of the papers were added next to each author.

- In the Results section, clarity could be improved by using separate subheadings for each type of cancer, rather than bullet points or numbers.

- Regarding brain tumors, PSMA PET has demonstrated strong uptake in high-grade gliomas, particularly GBM. Given the limited treatment options for GBM, incorporating PSMA PET imaging in a theranostic approach along with 177Lu-PSMA treatment could present a promising avenue. Could this be included in the paragraph? Moreover, a similar concept applies to patients with iodine-refractory thyroid cancer, where the lack of effective treatments could make them suitable candidates for 177Lu-PSMA treatment. Could this aspect also be addressed?

- On page 11, line 347, would adding "clear" before "superiority" soften the sentence?

- On page 13, line 447: the Ga-P16-093 tracer may not be familiar to the reader. Please consider describing it as a PSMA ligand.

- For each tumour, the authors effectively conclude with a summary, which enhances clarity. However, this appears to be missing for the RCC section. Could it be added?

- On page 14, line 515: please include in the sentence “the future of molecular imaging and treatment of tumours other than Pca…”

Author Response

Dear Guest Editors,

Please find enclose a copy of the revised manuscript in track-change function, entitled “PSMA PET Oncological Applications Beyond Prostate Cancer in Comparison to Other Radiopharmaceuticals”.

We would like to thank the reviewers for carefully and thoroughly reading our manuscript and providing valuable comments and suggestions that greatly improved the quality of our work.

Here we provide a point-by-point response:

 Reviewer 1)

The authors provide a comprehensive systematic review of PSMA PET oncological applications beyond prostate cancer. The review is meticulously written, and the paper is clear and concise. Moreover, the topic is interesting and relevant. I recommend the publication of the paper after a minor revision.

Q1. Table 1: Please provide a separate legend for opening the acronyms of the primary tumors. Moreover, it would improve clarity if reference numbers of the papers were added next to each author.

 R1. We thank Reviewer #1 for his valuable comments and suggestions. We have now modified Table 1 as requested. Specifically, we have defined the primary tumors without using abbreviations, and we have added a separate legend to explain the abbreviations used in the table. We have also added the reference numbers next to each author for clarity.

Q2: In the Results section, clarity could be improved by using separate subheadings for each type of cancer, rather than bullet points or numbers.

 R2. Following reviewer #1 suggestion, we have used separate subheadings for each type of cancer (despite the numbers) to improve the clarity of the results.

Q3. Regarding brain tumors, PSMA PET has demonstrated strong uptake in high-grade gliomas, particularly GBM. Given the limited treatment options for GBM, incorporating PSMA PET imaging in a theranostic approach along with 177Lu-PSMA treatment could present a promising avenue. Could this be included in the paragraph? Moreover, a similar concept applies to patients with iodine-refractory thyroid cancer, where the lack of effective treatments could make them suitable candidates for 177Lu-PSMA treatment. Could this aspect also be addressed?

R3. Thank you for this comment. We fully agree with the reviewer regarding the potential application of the treatment in the aforementioned patient population. Following your suggestion, we have added a comment on this in the Discussion section (lines 529-531, page 15). However, given the available evidence on 177Lu-PSMA treatment in tumors other than PCa, we have not emphasized this point at all because of the associated limitations, which are also mentioned in the manuscript. (Lines 527-535, page 15)

Q4. On page 11, line 347, would adding "clear" before "superiority" soften the sentence?

R4. We agree with the reviewer's observation. We now rephrase the sentence as follows:

“"Thus, according to the literature, PSMA expression in advanced HCC can be detected by [68Ga]Ga-PSMA-11 PET/CT, but without clear superiority over [18F]FDG PET/CT. Nevertheless, its potential utility as a complementary tool to MRI for T-staging of HCC should be further investigated.”

Q5. On page 13, line 447: the Ga-P16-093 tracer may not be familiar to the reader. Please consider describing it as a PSMA ligand.

R5. Following the reviewer's suggestion, we have now clarified this information in the paragraph (page 13, line 463)

 Q6. For each tumour, the authors effectively conclude with a summary, which enhances clarity. However, this appears to be missing for the RCC section. Could it be added?

R5. Thanks for the comment, we are sorry for the missing detail. We have added a summary sentence also in the RCC section.

Q7. On page 14, line 515: please include in the sentence “the future of molecular imaging and treatment of tumours other than Pca…”

R7. Done

Reviewer 2 Report

Comments and Suggestions for Authors

General comment: please use the nomenclature rules for radiopharmaceuticals as published by Coenen et al 10.1016/j.nucmedbio.2017.09.004 e.g. [68Ga]-labeled should be 68Ga-labeled

Please change unspecific names of radiotracers [68Ga]Ga-PSMA to the complete name e.g. [68Ga]Ga-PSMA-11  within the manuscript to make clear which radiopharmaceutical has been used in the study.

Page 1, line 78: [18F]PrhSMA-7.3 should read [18F]rhPSMA-7.3 

Comments on the Quality of English Language

Quality of English is fine but could be improved by language editing or review by a native speaker.

Author Response

Dear Guest Editors,

Please find enclose a copy of the revised manuscript in track-change function, entitled “PSMA PET Oncological Applications Beyond Prostate Cancer in Comparison to Other Radiopharmaceuticals”.

We would like to thank the reviewers for carefully and thoroughly reading our manuscript and providing valuable comments and suggestions that greatly improved the quality of our work.

Here we provide a point-by-point response:

Reviewer 2)

Q1. General comment: please use the nomenclature rules for radiopharmaceuticals as published by Coenen et al 10.1016/j.nucmedbio.2017.09.004 e.g. [68Ga]-labeled should be 68Ga-labeled

R1. We would like to thank reviewer #2 for the thorough reading of our manuscript and for the suggestions that have improved the quality of our work.

Q2. Please change unspecific names of radiotracers [68Ga]Ga-PSMA to the complete name e.g. [68Ga]Ga-PSMA-11  within the manuscript to make clear which radiopharmaceutical has been used in the study.

R2. We revised the whole manuscript accordingly.

Q3. Page 1, line 78: [18F]PrhSMA-7.3 should read [18F]rhPSMA-7.3 

R3 Done

Q4. Comments on the Quality of English Language: Quality of English is fine but could be improved by language editing or review by a native speaker.

R4. Thank you for your suggestion. We now extensively improved the whole manuscript English.

Reviewer 3 Report

Comments and Suggestions for Authors

Dear Authors,

Congratulations on your work.

Considering that I see no issues related to your manuscript and also research, I recommend it for publication.

Author Response

Dear Guest Editors,

Please find enclose a copy of the revised manuscript in track-change function, entitled “PSMA PET Oncological Applications Beyond Prostate Cancer in Comparison to Other Radiopharmaceuticals”.

We would like to thank the reviewers for carefully and thoroughly reading our manuscript and providing valuable comments and suggestions that greatly improved the quality of our work.

Here we provide a point-by-point response:

Q1. Dear Authors,

Congratulations on your work.

Considering that I see no issues related to your manuscript and also research, I recommend it for publication.

A1. Dear Reviewer #3, thank you very much for your supportive words and comments.